# Early Detection of Hearing Loss among the Elderly

**DOI:** 10.3390/life14040471

**Published:** 2024-04-04

**Authors:** Sol Ferrán, Raquel Manrique-Huarte, Janaina P. Lima, Carla Rodríguez-Zanetti, Diego Calavia, Constanza Jimena Andrade, David Terrasa, Alicia Huarte, Manuel Manrique

**Affiliations:** Clínica Universidad de Navarra, 31008 Pamplona, Spain; sferran@unav.es (S.F.); rmanrique@unav.es (R.M.-H.); crodriguezzanetti@unav.es (C.R.-Z.); dcalaviag@unav.es (D.C.); candrade@unav.es (C.J.A.); dterrasa@unav.es (D.T.); ahuarte@unav.es (A.H.); mmanrique@unav.es (M.M.)

**Keywords:** aging, hearing, prevention, early diagnosis

## Abstract

Background: Age-related hearing loss (ARHL) is a complex communication disorder that affects the cochlea and central auditory pathway. The goal of this study is to characterize this type of hearing loss and to identify non-invasive, inexpensive, and quick tests to detect ARHL among elderly adults, seeking to preserve quality of life and reduce the burden on healthcare systems. Methods: An observational, prospective study is conducted with >55-year-old subjects divided into the following groups: normal range (Group A), detected but not treated (Group B), and detected and treated (Group C). During follow-up, Speech Spatial Qualities (SSQ12), and Hearing Handicap Inventory in the Elderly Screening test (HHIE-S) questionnaires were assessed, along with hearing levels (hearing thresholds at 4 kHz were studied in more depth), and a series of tests and questionnaires to assess balance, cognitive level, level of dependence, and depression. Results: A total of 710 patients were included in this study. The duration of hearing loss (11.8 yr. in Group B and 21.0 yr. in Group C) and average time-to-treatment for Group C (14.1 yr.) are both protracted. Both of the used questionnaires show statistically significant differences among the groups, revealing greater handicaps for Group C. Audiometry performed at 4 kHz shows how hearing loss progresses with age, finding differences between men and women. There is a correlation between time-to-treatment in Group C and the cognitive test DSST (−0.26; *p* = 0.003). Conclusions: HHIE-S, SSQ12, and 4 kHz audiometry are sensitive and feasible tests to implement in screening programs.

## 1. Introduction

ARHL is a relevant problem, given its prevalence and morbimortality. Roth [1] and Stevens [2] note that 30% of European men and 20% of European women suffer from a 30 dB (or greater) hearing loss by the age of 70, as well as 55% of men and 45% of women aged 80 years. Therefore, central presbycusis increases by 4–9% per year (starting at around 55 years of age) and is more prevalent among men [3]. Hearing loss is disabling in approximately one-third of elderly adults in Europe. For example, it is estimated that 900,000 people suffering from hearing loss could be treated with a cochlear implant. This is a severe disorder in terms of morbimortality, with a huge functional impact on elderly adults, as it entails difficulties in communicating with others. It bears clear social, emotional, and health impacts, since it predisposes to falling [4,5,6], stress, anxiety and depression [6,7,8], and social isolation [9], and it fosters cognitive impairment and dementia [10,11,12]. All this creates a negative financial burden for the individuals, their families, and institutions, given the lack of income and employment this generates [13].

The early detection of hearing loss in newborns is a well-established concept at present, widely implemented in a significant number of geographical areas [14,15,16,17]. However, there is no experience in applying this type of program to the early detection of acquired hearing loss in adults. Screening programs have been developed based on questionnaires or mobile apps [18,19], but there is no systematic development, much less one implemented across healthcare systems.

Age-related hearing loss, also known as presbycusis, is a gradual hearing loss most people suffer as they age. This process is often exacerbated by intrinsic or extrinsic factors. ARHL is a relevant communication disorder that entails alterations to the cochlea and the central auditory pathway, eventually leading to difficulties in understanding spoken language [20]. Even with sufficient auditory sensitivity or audibility, understanding complex patterns of acoustic stimuli (language, music), particularly in a noisy environment [21,22,23] becomes arduous. Central neural processing speed and afferent integration time are disrupted. Moreover, inhibitory control and spatial memory have been seen to be lost as a result of the loss of sensory cells (hair cells) and progressive auditory deafferentation [24]. 

Unfortunately, ARHL is considered by many people today as an irremediable natural condition that will be suffered sooner or later, and there is a tendency to be passive towards it. There is a general perception that the longer one can hold out without aids, the better. However, delaying the time to treatment is actually detrimental to their health. Central ARHL must be perceived as an underestimated factor that accounts for broken interpersonal communications among the elderly and is coupled with other “non-auditory features”, such as balance disorders, falls, social isolation, depression, and cognitive impairment [25], severely affecting the quality of life among the elderly [26].

Early detection and diagnosis of hearing loss in the elderly would allow for early intervention, which would enable this population’s cognitive and mental skills and autonomy to be preserved [27]. This would improve their quality of life, reduce the negative impact that greater dependence would have on their caregivers, and enhance the sustainability of healthcare systems. 

The main objective of this study is to identify accurate, non-invasive, and rapid screening tests for the early detection of age-related hearing loss. These tests must offer high sensitivity and specificity; be non-invasive, inexpensive, quick to perform, and applicable in universal hearing screening programs in primary healthcare centers or other healthcare facilities and even autonomously, via mobile applications. 

## 2. Materials and Methods

### 2.1. Study Design

This is an observational study, including prospective measures to evaluate the effect of aging on hearing. Liminar tone audiometry and speech audiometry in quiet and in noise are the gold standard to assess the sensitivity and specificity of audiometries and easily administered questionnaires that could be used for the universal ARHL screening of elderly adults.

This study is observational as no additional intervention is applied to the treated subjects. Outcomes from routine practice are recorded through observational measures using standard clinical scales used widely in geriatrics and audiology. 

This study is part of the project “Hearing and balance in healthy aging”. The project was approved by the Clinical Research Ethics Committee of the Clínica Universidad de Navarra with file number 2017.174.

### 2.2. Population

Subjects were recruited at the otolaryngology department of the Clínica Universidad de Navarra and through an advertisement in a local newspaper requesting collaboration.

The following three subgroups of the population were studied for this purpose:-Group A comprised individuals aged 55 years and older with normal hearing and balance abilities, characterized by a pure-tone average of 0 to 20 dB across frequencies of 500, 1000, 2000, and 4000 Hz, and demonstrating good static and dynamic balance control.-Group B consisted of individuals aged 55 years and older who have been diagnosed with a hearing and/or balance disorder but have not undergone treatment, regardless of the reason.-Group C included individuals aged 55 years and older who have been diagnosed with a hearing and/or balance disorder and have received treatment: with hearing aids (HA), active middle ear implants (AMEIs), bone conduction implants (BCIs), cochlear implants (CIs), or vestibular rehabilitation.

#### 2.2.1. Inclusion Criteria

-The inclusion criteria for Group A required individuals to be aged 55 years or older, possess normal hearing in both ears, exhibit normal balance, demonstrate fluency in the Spanish language for clinical assessment, express willingness to engage in and adhere to all study protocols, and have the capacity to independently consent to participation.-The inclusion criteria for Group B mandated individuals to be aged 55 years or older, present with abnormal hearing in either ear and/or abnormal balance without receiving appropriate treatment, demonstrate fluency in the Spanish language for clinical evaluation, demonstrate a willingness to engage in and adhere to all study procedures, and have the capacity to independently consent to participation.-The inclusion criteria for Group C specified the inclusion of individuals aged 55 years or older who are unilateral or bilateral users of HA-BCI-AMEI-CI devices with bilateral hearing loss, received treatment when aged 55 years or older, meet the criteria for hearing aid treatment in one ear and for BCI-AMEI-CI treatment in the other ear, express a willingness to engage in and adhere to all study protocols, demonstrate fluency in the Spanish language for clinical evaluation, and have the capacity to independently consent to participation.

#### 2.2.2. Exclusion Criteria

The exclusion criteria included individuals who are significantly or severely reliant on others or are fragile; incapable of personally granting consent; unable to independently complete self-assessment questionnaires; have ossification or other cochlear anomalies impeding full electrode insertion in the case of cochlear implants (CIs); possess retrocochlear or central causes of hearing impairment; exhibit substantial comorbidities hindering participation in this study (such as blindness, immobility, severe aphasia, etc.); or have experienced clinical treatment failure related to chronic depression, dementia, or cognitive disorders.

Conditions such as sudden, idiopathic, or post-accident hearing loss were not excluded from this study as these will result in more severe hearing impairment than physiological presbycusis and will have a greater impact on the concept of aging on a global scale.

### 2.3. Study Measures: Assessment Tools

Anamnesis. Patients were asked about anamnesis and clinical history. Special attention was paid to establishing the duration of hearing loss, defined as the time elapsed between the first symptoms of hearing loss until the visit to the doctor’s office. Time-to-treatment was also accounted for in Group C.

The hearing assessment included the following tests—some are the gold standard, whereas others were included for their experimental value: -Unassisted pure-tone air-conduction audiometry (PTA): This assessment was conducted in soundproof chambers using the Interacoustics brand audiometer, model AC40. Unaided hearing thresholds, measured in decibel hearing level (dBHL) for pure-tone stimuli via air conduction, were assessed using headphones (TDH-39) following standard clinical procedures. Frequencies ranging from 250 to 4000 Hz were evaluated (250–500–1000–2000–4000 Kz), and a pure-tone average was calculated. This routine assessment typically takes around 5 min and involves both the clinician and the participant.-Speech discrimination assessment in quiet: This assessment was carried out in the participant’s typical listening environment, whether aided binaurally, aided monaurally, unaided in one ear, or unaided bilaterally (i.e., without hearing aid and cochlear implant availability). Standard speech materials were presented at various intensity levels in the sound field to establish the score–intensity function curve, including the presentation of a 65 dB sound pressure level (SPL). Recorded speech stimuli were presented from a loudspeaker at 00 Azimuth, at a distance of 1 m at head level. Disyllabic word lists in Spanish were utilized for this assessment. Results included the percentage of correctly identified speech items at 65 dB SPL and the speech reception threshold level (SRT50%). This evaluation typically lasts 10 to 15 min and involves both the clinician and the participant.-Routine speech discrimination assessment in noise: This evaluation involves assessing the ability to recognize speech in noisy environments, reflecting the subject’s everyday listening scenario, such as with bilateral hearing aids, one aided ear plus one unaided ear, or entirely unaided (without a hearing aid or cochlear implant). This evaluation is typically conducted for individuals who can accurately comprehend at least 50% of speech in quiet conditions. Testing occurs within a soundproof booth with added background noise. Depending on local protocols, speech materials are either presented adaptively in the sound field to determine the speech reception threshold (SRT) at 50% intelligibility in noise or fixed at a 65-decibel sound pressure level (dBSPL) while varying the background noise (e.g., pink noise) across presentation lists to ascertain the SRT at 50%. Speech samples and background noise are emitted from a speaker located directly ahead at a distance of 1 m from the subject’s head level. The test content typically comprises sentences or words commonly encountered in everyday listening situations in the native aided binaurally. This standard assessment procedure is typically conducted collaboratively by the clinician and the individual being assessed, lasting approximately 10 to 15 min.-Speech Spatial Qualities (SSQ12): SSQ12 serves as a self-evaluation tool for gauging hearing prowess and communication effectiveness in everyday settings, which is administered to patients [28]. It comprises twelve queries categorized into the following three sections: speech comprehension, spatial hearing (perception of sound in space), and quality (clarity of speech and other sounds), suitable for adults of all age groups and children aged nine and above [29]. Each query employs a rating scale ranging from 0 to 10, with higher scores indicating better proficiency. Typically, the scores are presented as mean ratings for each section, though they can also be analyzed individually or grouped differently, allowing for comparison between two time points. Clinically relevant disparities are identified by a rating shift of 1.0 for each subsection between testing sessions, a commonly observed discrepancy in evaluations of both unaided and aided hearing aid or implant users [30]. The completion of this form generally takes around 10 min for the candidate or recipient. Once completed, the questionnaire was reviewed by an audiology specialist.-Hearing Handicap Inventory in the Elderly Screening test (HHIE-S): The HHIE-S test is a concise self-evaluation tool tailored to measure the emotional and social impact of hearing loss on the daily lives of older individuals, both before and after receiving hearing-related interventions. It consists of ten questions, split evenly between emotional and social/situational aspects [31]. The total HHIE-S score ranges from 0 (minimal) to 100 (maximum). Scores below 16 in each subsection indicate no significant handicap, while scores falling between 17 and 42 denote a mild to moderate level of handicap, and scores exceeding 43 signify severe handicap. A higher HHIE-S score correlates with a greater degree of handicap caused by hearing impairment. Typically, candidates or recipients can complete this assessment in approximately 5 min. Once completed, the questionnaire was also reviewed by an audiology specialist.-Digit Symbol Substitution Test (DSST): The DSST is a neurological assessment tool renowned for its sensitivity in detecting brain injury, aging-related cognitive decline, and symptoms of depression, while also evaluating working memory [32]. It comprises a series of digit–symbol pairs (e.g., 1/-, 2/┴, … 7/Λ, 8/X, 9/=) followed by a list of digits. The task for the subject is to match each digit with its corresponding symbol as swiftly as possible. The score is determined by the number of correct symbol matches achieved within the allotted time frame, typically 120 s. Notably, a decline in symbol copying performance exhibits a robust correlation with advancing age. This assessment typically requires around 2 min to complete and is usually administered by a clinician alongside the candidate or recipient. Within the Wechsler Adult Intelligence Scale, this test is referred to as “Digit Symbol” (WAIS-R) or “Digit–Symbol-Coding” (WAIS-IV). An audiology specialist helped the subjects to perform this test.

It is important to clarify that, in the planning of the audiometric tests to be utilized, we estimated that the inclusion of the frequencies studied in tonal audiometry (250, 500, 1000, 2000, and 4000 Hz) was sufficient to obtain relevant information about the state of the peripheral auditory system. Additionally, we incorporated tests of verbal audiometry in both quiet and noisy environments, as well as questionnaires, to gather information about the state of central processing in the auditory pathway, which is also a significant aspect in the study of ARHL.

### 2.4. Statistical Analysis

A comparative study was completed for non-paired samples (hearing results and questionnaires) among the three study groups. Differences between categorical data were checked using the Chi2 test. For continuous data, when 3 groups were compared, the ANOVA test was used. In case of statistically significant differences, the differences between groups were checked using pairwise comparisons adjusted with Fisher’s LSD method. When 2 groups were compared, Student’s T test was used. 

Correlations between variables were performed using Spearman’s rank correlations due to non-multivariate normality.

In order to obtain sensitivity and specificity of our data over hearing loss, a ROC analysis was performed. The optimal cut point was set as the Youden index (furthest point from the diagonal). With the aim to check the merged effect of SSQ12, HHIE, and 4 kHz PTA, the same ROC analysis was carried out over the linear predictions of a multiple logistic regression model.

Distributional assumptions like the normal distribution were checked using the Shapiro–Wilk test and, due to the big sample size, graphic checks (Quantile-Quantile plots and Box plots) were used in order to check the real distribution. Homoscedasticity was checked using Levene’s test. Multivariate normality was checked using the Doornik–Hansen omnibus test.

IBM SPSS Statistics 20.0 program was used for data collection and analysis. A *p*-value less than 0.050 was considered statistically significant. The database created in this program included information on the demographic and clinical features and the results of the different tests and questionnaires.

## 3. Results

At the time of the last review in August 2023, 710 patients were included in this study. The distribution of patients was as follows: Group A—210 patients (128 women), Group B—302 patients (166 women), and Group C—198 patients (88 women). 

Below, we describe the demographic and audiometric data, as well as the auditory questionnaires used, in each of the three subgroups of the studied population. This serves as a starting point for conducting an analysis of results aimed at determining which of them offer high sensitivity and specificity and are non-invasive, inexpensive, quick to perform, applicable in universal hearing screening programs in primary healthcare centers or other healthcare facilities, and even autonomously, via mobile applications, which are the main objectives of this work. Additionally, an analysis is conducted on the impact that the duration of hearing loss and the time elapsed until treatment has on cognition and the success of auditory rehabilitation, whether with hearing aids or auditory implants.

The demographics and medical history of each group are listed in Table 1, from “Hearing and balance in healthy aging” [33].

Regarding demographic traits, the sole noteworthy distinctions noted among the three cohorts pertained to age, gender, educational attainment, bilingualism, and occupational engagement. Group C exhibited a greater proportion of males, with an average age of 68 years, in contrast to 66 and 61 years for Groups B and A, respectively. Within Group C, there was a reduced percentage of individuals with university education (28%) and lower rates of bilingualism (23%). Consistent with expectations, the employment rate was lower in Group C (29%) due to their retirement status.

### 3.1. Auditory Profile of Each Population Group 

Figure 1 depicts the clinical and audiometric characteristics of the hearing loss for Groups B and C based on the following criteria: unilateral or bilateral, sudden or progressive onset, type (conductive, sensorineural, and mixed). Bilateral, progressive, and sensorineural hearing losses were predominant.

This information was complemented with auditory thresholds in the left and right ear tone audiometry for each group and speech discrimination in quiet and in noise (Hint test). The results are plotted in Figure 2 and listed in Table 2 [33].

The average duration of hearing loss in Groups B (11.8 yr.) and C (21.0 yr.), as well as the average time-to-treatment for Group C (14.1 yr.), are plotted in Figure 3. Hearing loss was prolonged in both groups, in the range of years, and the results are widely diverse, with a greater average duration in Group C for which the average time-to-treatment (14.1 yr.) was greater than the duration of hearing loss in Group B (11.8 yr.).

### 3.2. Results of Questionnaires

Table 2 illustrates the total HHIE-S scores. Groups B and C exhibit significantly different values with respect to normality (A). However, only subjects from Group C scored above the normal range. This figure presents the results of the SSQ12 questionnaire as well, which indicates a statistically significant difference across the three groups, with Group C showing scores signifying a more severe handicap (*p* < 0.001).

### 3.3. Correlation between Time-to-Treatment in Group C and Audiometry and Questionnaire Outcomes 

We wanted to verify the impact of late treatment on audiometric tests and several questionnaires. There were no statistically significant correlations between time-to-treatment and speech audiometry (−0.06 (−0.23; 0.11), *p* = 0.471), the HINT test (0.02 (−0.15; 0.19), *p* = 0.784), or the SSQ12 (−0.04 (−0.21; 0.13), *p* = 0.637). Nevertheless, there was a significant correlation with the DSST cognitive test (−0.26 (−0.41; −0.09) *p* = 0.003), which suggests that the longer it takes to treat a patient, the worse the cognitive results.

### 3.4. Correlation between Standard Audiometries, Questionnaires, and Liminar Tone Audiometry at 4 kHz

Performing tone and speech audiometries, in quiet and in noise, are the gold standard to assess a person’s ability to hear. A statistically significant correlation (*p* < 0.001) was proven between each of the HHIE-S and SSQ12 questionnaires and the tone and speech audiometries, whether in quiet or in noise, for each group studied, A, B, and C (Figure 4).

The air conduction threshold at 4 kHz was analyzed separately. Using Groups A and B as a reference, Figure 5 displays the results of the average air conduction threshold for frequencies of 0.5 to 4 kHz, distributed by gender and age. The same figure exhibits the same analysis for a frequency of 4 kHz. These results reveal differences depending on gender, i.e., males hear worse. Similarly, it shows the progressive hearing impairment that comes with age. The results also point to the auditory threshold at 4 kHz as a predictive marker of the hearing impairment to come at other frequencies. This could be used in hearing loss detection tasks.

When analyzing the results of the 4 kHz frequency audiometry separately (Figure 5), gender differences persist to the detriment of men and hearing impairment progresses for years.

### 3.5. Sensitivity and Specificity 

Given the relationship between the three variables (HHIE, SSQ12, and audiometry at 4 kHz) detected in our data and the potential ability to detect hearing loss, we verified the predictive value of these variables with ROC curves, obtaining a cut-off point for each value with the Youden index.

Despite all variables being highly precise (AUC > 0.7), SSQ12 (AUC: 0.76 (0.73; 0.79)) performed worse, with 77.3% (74.2; 80.4) sensitivity, 63.9% (60.4; 67.5) specificity, 83.8% (81.1; 86.5) PPV, and 53.8% (50.2; 57.5) NPV followed by the HHIE questionnaire (AUC: 0.82 (0.79; 0.85)), with 86.6% (84.1; 89.1) sensitivity, 63.9% (60.4; 67.5) specificity, and 85.3% (82.7; 87.9) PPV 66.5% (63.0; 70.0), and finally, the audiometry at 4 kHz (AUC: 0.92 (0.90; 0.93)) with 79.7% (77.6; 81.8) sensitivity, 90.1% (88.6; 91.7) specificity, 95.1% (94.0; 96.2) PPV, and 64.8% (62.3; 67.2) NPV.

In light of these data, a combination of all these variables could comprise a useful model to screen for hearing loss quickly. Once all variables were collected in a multiple logistic model, the AUC for this model was 0.95 (0.93; 0.96), with 90.2% (88.0; 92.4) sensitivity, 86.4% (83.9; 88.9) specificity, 94.2% (92.4; 95.9) PPV, and 78.4% (75.4; 81.4) NPV. 

## 4. Discussion

The World Health Organization (WHO) promotes active aging, defined as “the process of optimizing opportunities for health, participation and security in order to enhance quality of life as people age. It allows people to realize their potential for physical, social, and mental well-being throughout the life course”. Furthermore, the World Health Assembly, a WHO body, in its resolution of 30 May 2017 [34], urged governments to add, among other things, strategies for otological and audiological care within the framework of primary healthcare systems and apply prevention and screening programs to the most exposed populations. 

In our sample of patients, the prevalence of hearing loss is 70.4% (50.8% women). It is normal to find a higher number of patients with hearing loss in our sample than in the general population since it can be expected that a study on hearing loss in older adults would involve more subjects affected by this condition than healthy ones. 

Detecting hearing loss in elderly adults is in line with the two goals mentioned by the WHO and meets the recommended criteria for universal screening [35]: it is a significant problem in terms of prevalence and morbi-mortality, with a known natural history, and a long and (latent) subclinical or presymptomatic phase. Regarding diagnosis, there must be acceptable tests for the population that are reliable (high positive predictive value, sensitivity, and specificity), simple, harmless tests with well-established and agreed-upon diagnostic criteria, as well as effective and available treatment. 

Current methods for detecting hearing loss include the scale method, subjective testing, and objective testing, such as the Hearing Handicap Inventory for Elderly Adults (HHIE), screening for otologic functional impairments (SOFI), pure tone audiometry (PTA), the subjective faces scale, the whispering experiment, speech audiometry, otoacoustic emissions, etc. However, the updated evidence report and systematic review from the U.S. Preventive Services Task Force showed that although some hearing loss screening tools have been developed, none of them have sufficient sensitivity, specificity, or positive and negative predictive values [36]. 

The results obtained with the HHIE, SSQ12, and pure tone audiometry at 4 kHz in our sample show high accuracy for all three tests, with a sensitivity > 77% in all of them, with the SSQ12 questionnaire being the least sensitive and the HHIE questionnaire the most sensitive. However, as we showed above, the combination of the three achieves a sensitivity of 90.2% and a specificity of 86.4%.

A study on the prevalence of hearing aid use among older adults in the United States showed that among people with hearing loss, the percentage of hearing aid users is 4.3% in individuals aged 50 to 59 years and 22.1% in individuals aged 80 years and older [37]. In our sample, of the 500 patients with hearing loss, 60% were not receiving any treatment.

Hearing aids and cochlear implants are effective, available treatments in a therapeutic approach for hearing loss in old age [38,39]. Some studies, including systematic reviews and a randomized controlled trial, have proven that hearing aids are a cost-effective intervention [40,41,42,43] that has a positive impact on improving the quality of life among their users, indicating significant improvements in various aspects of daily living, functional well-being, and emotional health [44,45,46]. Additionally, positive results have been reported by hearing aid users compared with non-users. They report improvement when socializing and better mental and physical health [46]. The use of hearing aids mitigates the risk of social dependence and premature death [47,48] and exerts a positive effect on depression [49]. There are increasingly more studies that prove that cognitive impairment can be reduced with the use of hearing aids [50]. Cochlear implants, given that they restore hearing, reduce the prevalence of tinnitus, improve quality of life, lessen symptoms linked to depression, and improve overall cognitive performance [51,52,53]. The cost–benefit analysis of cochlear implantation has been well-established by a series of systematic reviews and other research [54,55].

However, the results of this study point to an average of 11.8 years of untreated hearing loss in Group B and 14.1 years on average in Group C. As in children, these protracted periods of hearing loss lead to deafferentation of the auditory pathway and auditory centers and cause plastic changes in this and other areas connected to hearing, especially visual, somatosensorial, and frontal cortices. Campbell and Sharma [56] describe how these plastic changes occur at the onset of hearing loss in adults with mild hearing loss. 

Although hearing aids and cochlear implants have proven to be effective in treating hearing loss [38,39,40,41,42,43], these prolonged periods of hearing deprivation without treatment are conspicuous. The penetration estimation rate for hearing aids in patients who need them is around 10–11%, although in the group of high-income countries, as defined by the WHO, the coverage is 57% [57]. There are numerous reasons that account for this very low rate of the use of hearing aids including the following: the person affected is in denial; there is social stigma linked to hearing loss, which, as a result of aging, makes patients reluctant to wear a visible hearing aid or cochlear implant; and they have unrealistic expectations in many cases due to inept, unprofessional information. 

Hearing loss is often a silent burden. According to the 2007 “Hear the World” study, the extent to which people close to the patient are aware of their hearing problem varies: the closer the relation, the more aware they are, but still, 46% of family members are unaware. This number grows to 61% in the person’s social circle and 78% at work. In most cases, family and friends are more aware of the problem than the patient is. This explains why 43% of people with hearing loss have never tested their hearing. However, 46% of people with sight problems check their vision every year. 

On a different note, even though it would be recommendable for healthcare professionals to screen for one of the three most prevalent chronic diseases affecting elderly patients—hearing loss being one of them—this is not performed. Even once hearing loss is detected, doctors collect partial data, which reflects knowledge gaps in all features of hearing loss among the elderly [58].

These data reveal that a global approach is needed today more than ever to tackle the problem by raising awareness among healthcare professionals and wider society and implementing early detection, diagnosis, and intervention programs with optimal follow-up to reap the benefits sought.

The goal would be to detect hearing loss early among elderly individuals (>60 years) and then carry out a comprehensive, early intervention and promote good hearing, reducing the rate of cognitive alterations, dependence, and depression and fostering a high-quality, active lifestyle. 

Even though there are several methods to achieve this goal, our proposal for healthcare would be to detect hearing loss in individuals aged 60 years and older through primary healthcare centers. This study identified the HHIE-S and SSQ12 questionnaires as extremely useful tools for detecting hearing loss early in this population group. These tests are free, easy, and quick to perform. Therefore, they could be used in an early hearing loss detection program in primary healthcare centers. Patients could fill in the questionnaire(s) while in the waiting room, and following a simple correction by the nursing staff, the doctor would have data to then confirm the diagnosis through the Audiology Unit and start treatment with an ENT specialist if needed.

Furthermore, this research explores the possibility of using the air conduction threshold at 4 kHz as a simple method to detect ARHL, as it would simplify the tone audiometry analysis, bringing it closer to the requirements of a detection test. As previously mentioned, there are differences by gender, where hearing levels are poorer among men and hearing worsens progressively with age. This suggests that the hearing threshold at 4 kHz is a predictive marker for hearing impairment at other frequencies, thereby making it useful for detecting hearing loss. 

Moreover, based on the results of our work, there is a strong statistical correlation between the HHIE-S and SSQ12 questionnaires and the auditory threshold at 4000 Hz, as well as the outcome of standard audiometric tests, such as tone and speech audiometries in quiet and in noise.

Currently, with the development of digital technologies, there are applications that can gather audiometric data using various devices, such as smartphones, miniaudiometers, and tablets [57,59]. These devices could be used as an initial screening technique, by emitting a sound starting at 4000 Hz, at varying intensities, to detect the hearing threshold before referring the patient to the Audiology or ENT Unit. This methodology could be used in primary healthcare centers, other healthcare facilities, or by the subjects themselves. Because they are bloodless, easy to use, and effective enough, they can be used to diagnose hearing loss in elderly adults.

It must be noted that this study has certain limitations, such as, for example, that the sample came from a single center. If the subjects came from different centers, in addition to a larger number of patients, there may have been more variety in the demographic characteristics. It should also be considered that any self-reported data, as in the case of questionnaires, should be treated with more caution than an objective test.

## 5. Conclusions

The duration of hearing loss and time-to-treatment was extensive at more than 10 years. This long period of auditory deprivation has negative consequences. A negative correlation was established between the DSST cognitive test and the duration of hearing loss prior to treatment (−0.26; *p* = 0.003), which suggests that a delay in treatment predisposes to cognitive impairment. 

The HHIE-S and SSQ12 questionnaires and air conduction threshold at 4 kHz are quick, easy tests, not to mention being inexpensive and reliable, with high sensitivity and specificity. They can be implemented in hearing loss detection programs for adults.

Using the air conduction threshold of 4 kHz was profiled as a possible predictive marker for hearing loss, with differences based on gender and older age.

Although expectations with these results are promising, large multicenter implementation trials will be needed to confirm the efficacy and acceptability of the screening model in actual primary care. More research is needed in the future to implement this type of screening, such as examining adherence to screening referrals, the benefits/costs of early intervention, or the impact on long-term cognitive outcomes.

## Figures and Tables

**Figure 1 life-14-00471-f001:**
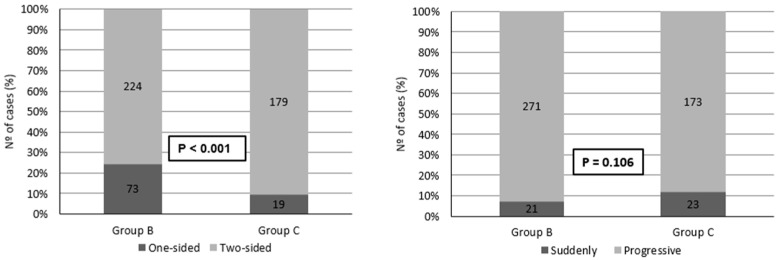
Clinical and audiometric traits of hearing loss profiles for Groups B and C.

**Figure 2 life-14-00471-f002:**
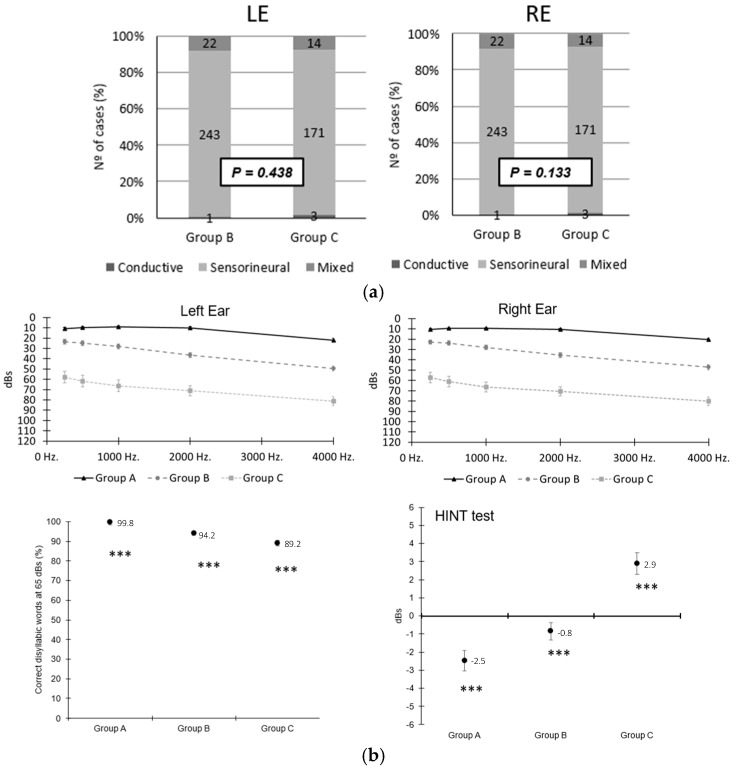
(**a**) Liminar tone audiometry for each group. (**b**) Speech tone audiometry, in quiet and in noise (Hint test) for each ear and group.

**Figure 3 life-14-00471-f003:**
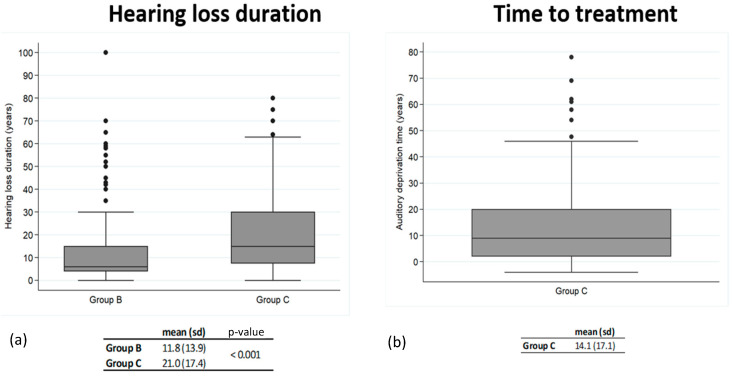
(**a**): Duration of hearing loss in groups B and C; (**b**): Time elapsed in group C until receiving treatment.

**Figure 4 life-14-00471-f004:**
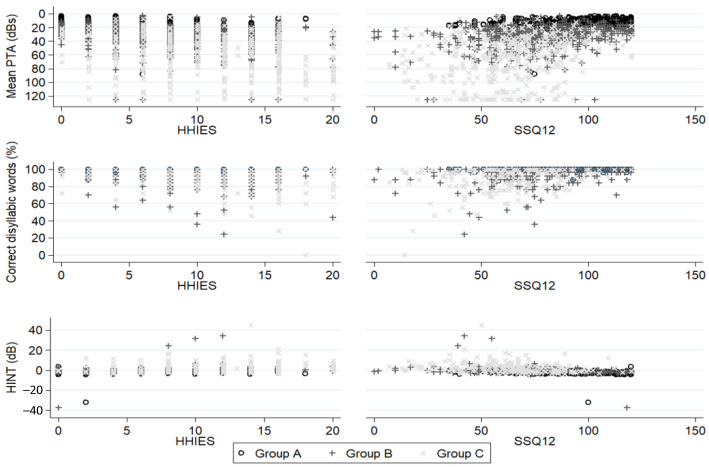
Correlation between standardized audiometric tests (tone and speech audiometry, in quiet and in noise, Hint test) and HHIE-S and SSQ12 questionnaire scores for each group.

**Figure 5 life-14-00471-f005:**
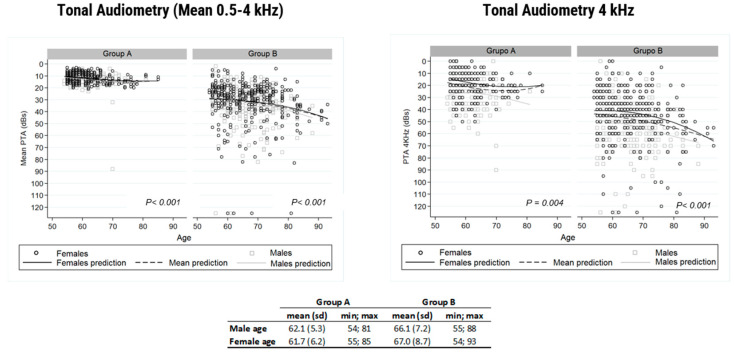
The figure to the right depicts the correlation between average auditory thresholds obtained with tone audiometry at frequencies between 0.5 and 4 kHz and age, in Groups A and B for men and women. The figure on the left reveals the same correlation; however, in this case, it considers auditory thresholds at 4 kHz.

**Table 1 life-14-00471-t001:** Demographics and medical history of each group.

Chracteristics	Group A (n = 210)	Group B (n = 302)	Group C (n = 198)	*p*-Value
Gender (M/F)	61% F	55% F	44% F	*p* < 0.010
Age	61 y	66 y	68 y	*p* < 0.001
Habitat (rural/urban)	83% urban	78% urban	73% urban	NSD
Assistance (none/family)	99% none	97% none	98% none	NSD
Education level	50% university	38% university	28% university	*p* < 0.001
Languages (mono/plurilingual)	66% monolingual	70% monolingual	77% monolingual	*p* < 0.050
Work activity	53% yes	34% yes	29% yes	*p* < 0.001
Tobacco	55% no	47% no	45% no	NSD
Cardiovascular disease	28% yes	31% yes	31% yes	NSD
Psychiatric disease	18% yes	21% yes	18% yes	NSD
Neurological disease	4% yes	8% yes	9% yes	NSD
Endocrine disease	31% yes	31% yes	32 yes	NSD
Mobility (normal/assisted)	1% assisted	2% assisted	5% assisted	NSD

F—female/M—male/y—years/NSD—no significant difference.

**Table 2 life-14-00471-t002:** The results of the hearing tests for each of the study groups.

PTA	500 Hz	1000 Hz	2000 Hz	4000 Hz	Total (dB)
Group A	9.7 dB (6.6) (0; 90)	9.3 dB (6.2) (0; 85)	10.3 dB (8.0) (0; 85)	21.2 dB (12.3) (0; 90)	12.6 dB (5.9) (2.5; 87.5)
Group B	24.2 dB (18.9) (0; 120)	28.0 dB (19.1) (0; 120)	35.9 dB (19.8) (0; 125)	48.4 dB (20.5) (0; 120)	34.1 dB (16.9) (1.2; 120)
Group C	61.4 dB (38.2) (0; 120)	66.4 dB (36.9) (5; 120)	70.9 dB (33.8) (0; 125)	80.8 dB (31.6) (0; 120)	69.8 dB (33.5) (3.7; 120)
**Speech Audiometry in Quiet**			**Speech Audiometry in Noise (Hint Test)**		
Group A	99.8% (1.1) (88; 100)		Group A	−2.5 dB (1.0) (−4.4; 3.2)	
Group B	94.2% (10.6) (24; 100)		Group B	−0.8 dB (3.6) (−4.2; 34.0)	
Group C	89.2% (14.1) (0; 100)		Group C	2.9 dB (5.5) (−3.5; 45.0)	
**HHIE-S Total**	**Total (Range 0–120)** 7.1 (5.7) (0; 20)		**SSQ12**	**Total (Range 0–120)** 78.9 (25.0) (0; 120)	
Group A	2.9 (3.8) (0; 18)		Group A	94.8 (19.7) (35; 120)	
Group B	7.3 (4.9) (0; 20)		Group B	78.3 (23.4) (0; 120)	
Group C	11.5 (5.7) (0; 20)		Group C	63.1 (22.1) (7; 120)	

Data are presented as mean (sd) (min; max).

## Data Availability

All data generated or analyzed during this study are included in this article. Further inquiries can be directed to the corresponding author.

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
