# Peer review of "Early Detection of Hearing Loss among the Elderly"

_life, 2024, doi:10.3390/life14040471_

Round 1

Reviewer 1 Report

Comments and Suggestions for Authors

The abstract needs improvement. It is quite chaotic and does not present the tools and results in a clear way. The reader does not get a clear message of who and what was studied and what results were obtained.

Line 76 first introduces the abbreviations HA, AMEI, BCI, CI, which those who are familiar with the subject will understand, but not every reader is expected to be familiar with it....

line 71 "in the better ear" does this mean that a person could have a hearing loss in one ear and with the other ear hearing well was classified in the "healthy" group? (later clariefied in 78 line... line 71 should be corrected)

Why was the criterion of imbalance introduced, when the title of the work projects the topic of hearing. Of course, I accept that these two aspects are connected, but the introduction as well as the abstract and title do not suggest an analysis of the aging processes of the inner ear, only presbyacusis on the quality of life of young seniors. In doing so, it would be worthwhile to refer to WHO standards since when we have been talking about aging and what early intervention should be in view of this.

If the premise of the study was to analyze early detection of hearing disorders in the elderly, shouldn't diabetes, cardiovascular disorders and cardiovascular diseases be included in the exclusion criteria?

Standard tonal audiometry shows pitch from 250 Hz to 8,000 Hz. Schuknecht and Gacek distinguish four forms of presbyacusis based on pathogenetic factors:
- sensory type - atrophy of the cells of the organ of Corti in the basal bend, hearing loss for treble tones
- neural type - impaired function of cochlear neurons and the auditory pathway
auditory pathway, poor speech comprehension,
- Metabolic type - impairment of biochemical processes in the auditory and nerve cells.
auditory and nerve cells, hearing loss affecting all
frequencies,
- mechanical type - lesions of an ischemic, mixed nature.
In view of this, why did the authors of the presented study omit from the analysis the frequencies of 6 and 8 kHz, which seem to be crucial for the analysis of the issue of hearing loss in seniors.

It is puzzling to take into account patients with CI, their ability to understand speech depends on the time of rehabilitation and use of CI, patients with "freshly" put on CI will have much worse results than those using it for many years. Also, a CI patient tested without a CI in a speech comprehension test will have a score of 0. Perhaps it would be useful to make this more specific?

The title of the paper indicates early detection of hearing loss in the elderly, in view of this I understand we are referring to presbyacusis. With this assumption, shouldn't people with sudden, idiopathic, or post-accident hearing loss be excluded from the analysis?

In the discussion, the authors emphasize tonal audiometry up to 4kHz as a diagnostic tool, where in fact the entire range of frequencies tested in PTA as the gold standard was not analyzed in the paper. Moreover, the authors discuss presbyacusis and include patients with sudden hearing loss in the material analyzed - the material and methods seem to need to be clarified and edited.

40% of the references is from the last 5 years.

Reviewer 2 Report

Comments and Suggestions for Authors

Abstract:

- Briefly state the study design, number of subjects, age range, and methods used.

- Summarize the key results such as prolonged untreated hearing loss, correlation between treatment delay and cognition, validity of questionnaires and 4KHz audiometry for screening.

- Concisely conclude that the HHIE-S, SSQ12, and 4KHz audiometry are sensitive and feasible tests to implement in screening programs.

Introduction:

- Provide more background on prevalence and impact of age-related hearing loss to establish significance.
- line 31, middle ear disease should be considered in differential diagnosis as confounding varaibles. cite doi:10.3390/jcm11237000
- Elaborate on current barriers leading to under-detection and long delays before treatment.

- early hearing loss is essential to allow better hearing outcomes. cite doi:10.1093/brain/awaa429.
- Clearly state the rationale and specific study aims to identify accurate, non-invasive, and rapid screening tests for early detection of age-related hearing loss.

Methods:

- Provide more details on participant recruitment - setting, inclusion/exclusion criteria, sampling method etc. This will help readers evaluate potential selection bias.

- Describe the standardized procedures followed for audiometric assessments like pure tone audiometry and speech tests. Specify equipment used and testing conditions.

- Elaborate on the administration of questionnaires - whether self-reported or administered by study staff; steps to minimize missing data.

- Explain statistical analysis plan - rationale for tests used, tools for checking assumptions, significance threshold etc. This enhances transparency.

Results:

- Focus results on key findings that connect back to study aims and hypotheses. Move secondary analyses to supplementary data.

- Include quantitative summary statistics like means, standard deviations for audiometric thresholds and questionnaire scores in each group. Adds precision.

- Use tables/figures judiciously to depict group differences in hearing profiles, duration of hearing loss etc. But avoid repetitive figures.

- While correlating hearing loss duration with outcomes, include effect sizes and confidence intervals to show magnitude of associations.

- Specify sensitivity, specificity, predictive values for proposed screening tests at optimal cut-offs identified. Strengthens clinical applicability

Discussion:

- Interpret key results in context of prior literature - compare hearing loss prevalence, treatment delays, and screening test accuracy to existing evidence. This enables meaningful interpretation.

- Acknowledge limitations like single-center sample, lack of causal inferences, self-reported data issues etc. to balance conclusions.

- a criticism in hearing loss diagnosis was the lack and delay of hearing screening during pandemic. Several patients avoid examinations. Please consider and cite doi:10.1007/s00405-021-06958-4

- Avoid overstating clinical implications until large multi-center implementation trials confirm efficacy and uptake of screening model in real-world primary care.  

- Propose future directions like examining adherence to screening referrals, benefits/costs of early intervention, impact on cognitive outcomes over long term. This highlights remaining knowledge gaps.

Conclusions:

- Briefly recap main findings - untreated hearing loss duration, screening tools validity, negative cognitive impact of delaying treatment etc.

- Emphasize need to confirm screening protocol value through controlled trials in diverse settings before wide adoption.

- Note study provides initial evidence and rationale to spur future research on early hearing loss detection in the elderly. But definitive practice recommendations are premature currently.

Comments on the Quality of English Language

english language need minor grammatical corrections

Round 2

Reviewer 1 Report

Comments and Suggestions for Authors

I accept all clarifications and corrections.